# Metabolomics Signatures and Subsequent Maternal Health among Mothers with a Congenital Heart Defect-Affected Pregnancy

**DOI:** 10.3390/metabo12020100

**Published:** 2022-01-21

**Authors:** Ping-Ching Hsu, Suman Maity, Jenil Patel, Philip J. Lupo, Wendy N. Nembhard

**Affiliations:** 1Arkansas Center for Birth Defects Research and Prevention, Fay W. Boozman College of Public Health, University of Arkansas for Medical Sciences, Little Rock, AR 72205, USA; SMaity@uams.edu (S.M.); Jenil.Patel@uth.tmc.edu (J.P.); Philip.Lupo@bcm.edu (P.J.L.); 2Department of Environmental and Occupational Health, Fay W. Boozman College of Public Health, University of Arkansas for Medical Sciences, Little Rock, AR 72205, USA; 3Department of Epidemiology, Human Genetics, and Environmental Sciences, School of Public Health, University of Texas Health Science Center at Houston (UTHealth), Dallas, TX 75207, USA; 4Department of Pediatrics, Section of Hematology-Oncology, Baylor College of Medicine, Houston, TX 77030, USA; 5Department of Epidemiology, Fay W. Boozman College of Public Health, University of Arkansas for Medical Sciences, Little Rock, AR 72205, USA

**Keywords:** metabolomics, maternal health, congenital heart defects

## Abstract

Congenital heart defects (CHDs) are the most prevalent and serious of all birth defects in the United States. However, little is known about the impact of CHD-affected pregnancies on subsequent maternal health. Thus, there is a need to characterize the metabolic alterations associated with CHD-affected pregnancies. Fifty-six plasma samples were identified from post-partum women who participated in the National Birth Defects Prevention Study between 1997 and 2011 and had (1) unaffected control offspring (n = 18), (2) offspring with tetralogy of Fallot (ToF, n = 22), or (3) hypoplastic left heart syndrome (HLHS, n = 16) in this pilot study. Absolute concentrations of 408 metabolites using the AbsoluteIDQ^®^ p400 HR Kit (Biocrates) were evaluated among case and control mothers. Twenty-six samples were randomly selected from above as technical repeats. Analysis of covariance (ANCOVA) and logistic regression models were used to identify significant metabolites after controlling for the maternal age at delivery and body mass index. The receiver operating characteristic (ROC) curve and area-under-the-curve (AUC) are reported to evaluate the performance of significant metabolites. Overall, there were nine significant metabolites (*p* < 0.05) identified in HLHS case mothers and 30 significant metabolites in ToF case mothers. Statistically significant metabolites were further evaluated using ROC curve analyses with PC (34:1), two sphingolipids SM (31:1), SM (42:2), and PC-O (40:4) elevated in HLHS cases; while LPC (18:2), two triglycerides: TG (44:1), TG (46:2), and LPC (20:3) decreased in ToF; and cholesterol esters CE (22:6) were elevated among ToF case mothers. The metabolites identified in the study may have profound structural and functional implications involved in cellular signaling and suggest the need for postpartum dietary supplementation among women who gave birth to CHD offspring.

## 1. Introduction

Congenital heart defects (CHDs) affect approximately 1% of all births in the United States [1,2,3]. These conditions represent a serious public health problem as they are a leading cause of death by disease in children [4,5,6], and those who survive often require repeated surgeries and hospitalizations. While understanding the etiologies of CHDs and outcomes among these children remains an important focus of research, much less is known about the outcomes among mothers who have children with these conditions.

This is particularly important, as disrupted metabolism related to the affected birth could impact long-term maternal health. Women with pregnancies affected by CHD have been reported with altered metabolic profiles and were associated with dysregulated metabolic pathways [7,8,9,10]. These markers were often utilized as predictive markers for fetuses with congenital heart defects. However, limited information is available on the mothers’ postpartum health outcome, which might influence not only the health of the mothers but also the children.

Pregnancy is characterized by substantial changes in metabolism and could influence subsequent health outcomes. Metabolomics is a relatively recent addition to omics-based platforms [11]. The biochemical pathways related to metabolic byproducts are measured via chromatographic or nuclear magnetic resonance platforms to estimate the effect of certain stressors in physiological systems [11,12,13,14,15,16]. It has been used to study the physiological impact of toxic exposures [17,18,19] and environmental pollutions [20,21,22,23].

Metabolomics has also been widely used to detect biomarkers in a variety of pathological conditions, including diabetes and insulin resistance [24,25,26], pediatric diseases [27,28], inborn errors of metabolism), cardiovascular diseases [29,30,31], cancers [32,33,34,35,36,37], and aging [38,39] and can serve as useful tools for drug development [40,41,42]. The change in the relative abundance of lipids, amino acids, or carbohydrates influenced by external and internal catalytic processes can offer valuable insights regarding progression of a disease or severity of their impact [43,44,45] and has the potential to predict maternal and perinatal health [46,47,48].

Recently, there have been important strides in improved detection thresholds, analyses, and linking biological pathways associated with diseases using metabolomics [49,50,51]. As they are likely to be an important biomarker of long-term maternal health, our objective was to characterize the metabolic alterations associated with CHD-affected pregnancies in this pilot study among post-partum women who participated in the National Birth Defects Prevention Study (NBDPS).

## 2. Materials and Methods

### 2.1. Study Design and Study Participants

Samples on case and control mothers were obtained from the National Birth Defects Prevention Study (NBDPS). Briefly, the NBDPS was a large case-control multicenter study of birth defects in the US, funded by the Centers for Disease Control and Prevention (CDC). The methods have been described previously [52]. Furthermore, subjects selected for this analysis were included in a subsequent follow-up study conducted in Arkansas (an NBDPS site) on maternal biomarkers and risk of offspring with CHDs [52].

Specifically, 550 mothers of cases with CHDs and 221 control mothers were recruited between March 2001 and June 2005. After receiving written informed consent, blood samples were obtained at least 6 weeks postpartum by standard venipuncture (up to 30 mL) and were immediately placed on ice and delivered to the laboratory for storage. Processed samples, including plasma, RBC, DNA, and urine, were aliquoted and stored in locked −80 °C freezers, which are only accessible to authorized study personnel and facility management.

The temperature of the freezer were continuously monitored by an electronic monitoring system and routinely monitored by the biorepository personnel to ensure the integrity of the samples. Prior to sample collection, participants were asked to refrain from eating for at least 3 h prior to the blood sample collection. For this assessment, we selected the following groups: (1) mothers of unaffected offspring (mControl, n = 18); (2) mothers of offspring with tetralogy of Fallot (mToF, n = 22); and (3) mothers of hypoplastic left heart syndrome (mHLHS, n = 16). These case groups were selected based on phenotypic severity. For this analysis, control mothers were matched to mothers of cases on age and race/ethnicity.

### 2.2. Targeted Metabolite Profiling

For the metabolomics assessment, 50 μL of plasma samples from 56 participants were used for the targeted metabolite profiling, and 26 out of 56 samples were randomly selected as technical repeats in the analytical run. Metabolomic profiling was conducted using a commercial reverse-phase liquid chromatography and tandem mass spectrometry (LC-MS/MS) kit (AbsoluteIDQ^®^ p400 HR Kit, Biocrates Life Science AG, Innsbruck, Austria), including isotope-labeled internal standards of 408 metabolites from 11 compound classes as well as quality control samples and reagents for the derivatization and extraction of metabolites using multiple reaction monitoring (MRM) ion pairs for metabolite identification and quantification.

The complete analytical process was conducted by the Metabolomics Innovation Centre (TMIC) at the University of Alberta. In brief, the separation of amino acids and biogenic amines was performed using Thermo Vanquish UHPLC with a C18 column (Biocrates, Part 9120052121032) and guard column (Biocrates, Part 9120052121049). Analytes were separated using a gradient from 0.2% formic acid in water to 0.2% formic acid in acetonitrile as indicated in the gradient table below. The total UHPLC analysis time was approximately 5.8 min per sample.

Acylcarnitines, monosaccharides (hexose), diglycerides, triglycerides, lysophosphatidylcholines, phosphatidylcholines, sphingomyelins, ceramides, and cholesteryl esters were analyzed by flow injection analysis (FIA) with a total analysis time of approximately 3.1 min per sample. Biocrates provided the FIA mobile phase buffer (Part 9120052121018), which was diluted into LC-MS grade methanol for use with the kit per the manufacturer’s instructions.

Using electrospray ionization in positive ion mode, samples for both UHPLC and flow injection analysis were introduced directly into Q Exactive™ Orbitrap MS systems operating in the full scan or parallel reaction monitoring (PRM) mode. Acquisition methods and tune parameters for all instruments were provided by Biocrates as part of the p400H kit. Data analysis was performed using MetIDQ provided by Biocrates.

### 2.3. Metabolomics Data Analysis

Metabolomics data were processed using MetaboAnalyst 4.0 [53]. Metabolites below the limit of detection (LOD) were replaced by a value half of the minimum peak intensity of the entire dataset. Quantile normalization was conducted to reduce sample-to-sample variation, followed by log^2^ transformation to ensure that the data followed the assumptions of normality and further by mean center scaling to diminish the error in multivariate analysis.

Principal component analysis (PCA) as well as hierarchical clustering were constructed using the Partek Genomics Suite (St. Louis, MO) to obtain a 2- and 3-dimensional visualization of the profiles. Analysis of covariance (ANCOVA) adjusting for maternal age at delivery and body mass index (BMI) with Fisher’s least significant difference contrast method was used to assess the differential metabolites univariately.

A significance level of *p* < 0.05 was used to define statistical significance. Based on the significant metabolites (*p* < 0.05), we developed a classification model using a logistic regression with the receiver operating characteristic curve (ROC curve) and the associated area under the curve (AUC) to estimate the predictive accuracy for differentiating between cases and controls. The flowchart describes the primary steps involved in the data-analyses (Figure 1).

### 2.4. Pathway Analysis

The pathway enrichment was performed using the set of significant metabolites from the following comparisons (1) mControls compared to mToF and (2) mControls compared to mHLHS. The HMDB IDs were mapped onto their corresponding KEGG IDs. The pathway enrichment followed the associated diffusion matrix threshold using the selection criteria of enrichment *p* < 0.01 with a minimum of 20 pathways and 250 individual nodes comprised of pathways, enzymes, compounds (metabolites), and associated reactions. Final list of pathways were manually rendered based on most relevant metabolic networks associated with the maternal health affected by the CHD pregnancies. Separate network graphs were generated for the two comparisons described using Cytoscape v.3.8.2.

## 3. Results

### 3.1. Characteristics of Study Participants

Participants in this study (n = 56) were predominantly of European descendent (91%, see Table 1). The mean age of study participants was 26.5 (range 16–36, SD = 5.5) years, with an average pre-pregnancy BMI of 25.8 kg/m^2^ (range 16.1–44.6 kg/m^2^, SD = 6.5), average post-partum BMI of 30.5 kg/m^2^ (range 21.3–46.9 kg/m^2^, SD = 5.8), and weight change of 12.7 kg (range 0–38.6, SD = 7.0). High blood pressure was observed among nine women. Notably, none of the variables were significantly different while comparing mHLHS and mToF with mothers of controls (Table 1).

### 3.2. Targeted Metabolomics Profiling

A total of 408 metabolites were assayed, with an average limits of detection (LOD) detailed in the Appendix A. Eight metabolites were not detected in all of our samples, including PC (30:3), PC-O (33:4), TG (56:9), acetyl-Ornithine, carnosine, dopamine, histamine, and phenylethylamine. Therefore, 400 metabolites were included in the final analysis. Summary of data processing results, including the number of metabolites imputed due to missing or below LOD value can be seen in Appendix A.

Twenty-six samples serving as technical repeats were randomly selected and distributed among all samples. The high correlation between samples and its repeats represented the reliability of the detection technology (Appendix A). The year of sample collection from 18 control samples ranged from the year 1999 to 2008 and did not show any mass spectrometer variations base on the PCA plot shown in Appendix A.

### 3.3. Maternal Metabolites Associated with CHD-Affected Pregnancies

PCA modeling and hierarchical clustering of the significant metabolites demonstrated separation of metabolomic profiles between mControl vs. mHLHS (Figure 2a,b). In the ANCOVA model, adjusting for maternal age at delivery and BMI (Appendix A), we identified nine metabolites that were significantly different between the mothers of cases vs. mothers of controls (*p* < 0.05).

Among those metabolites, PC (34:1), PC (41:4), SM (31:1), SM (42:2), and PC-O (40:4) were consistently higher among HLHS. TG (52:6), PC (41:1), glutamine, and PC (35:0) were lower among mHLHS than mControls (Table 2). In the assessment of ToF-affected pregnancies (Figure 3a,b), we identified 30 metabolites that were significantly different between mToF and control mothers (*p* < 0.05) using the ANCOVA model adjusting for maternal age at delivery and BMI (Appendix A). Among those metabolites, six were higher in mToF, whereas 24 were lower among mToF (Table 3). When comparing significant metabolites from both analyses, PC (35:0) was the only metabolite consistently lower among mHLHS and mToF women compared with mControls (Figure 4).

### 3.4. Biomarker Analysis

Based on the ROC analyses among mHLHS, four out of nine metabolites presented with higher classification potentials (AUC values > 0.70). These included: PC (34:1) (AUC: 0.74, 95%: 0.56–0.89); SM (31:1) (AUC: 0.72, 95%: 0.52–0.88); SM (42:2) (AUC: 0.70, 95%: 0.51–0.87); and PC-O (40:4) (AUC: 0.76, 95%: 0.59–0.90) (Figure 5a–d). For mothers of ToF cases, 5 out of the 30 significant metabolites presented with higher classification potentials. These include: LPC (18:2) (AUC: 0.71, 95%: 0.54–0.87); TG (46:2) (AUC: 0.71, 95%: 0.54–0.86); LPC (20:3) (AUC: 0.71, 95%: 0.52–0.85); CE (22:6) (AUC: 0.74, 95%: 0.57–0.89); and TG (44:1) (AUC: 0.73, 95%: 0.55–0.88) (Figure 6a–e).

### 3.5. Pathway Analysis

From the pathway analysis using the nine metabolites identified in the mHLHS analysis, 13 pathways were significantly projected to the network (*p* < 0.01, Figure 7a), with the D-glutamine and D-glutamate metabolism, sphingolipid metabolism, sphingolipid signaling pathway, glutamatergic synapse, and proximal tubule bicarbonate reclamation as the top pathways affected (*p* = 1 × 10^−6^). From the pathway analysis using the 30 metabolites identified in the mToF analysis, 18 pathways were significantly projected to the network (*p* < 0.01, Figure 7b) with the glutamatergic synapse, long-term depression, GnRH signaling pathway, pancreatic secretion, central carbon metabolism, choline metabolism, lipid and atherosclerosis, and acylglycerol degradation as the top pathways affected (*p* = 1 × 10^−6^).

## 4. Discussion

Overall, we found that mothers of children with CHDs had metabolomic profiles associated with phospholipid and glutamate metabolism, suggesting that CHD-affected pregnancies may have an influence on the long-term maternal metabolic health. This is particularly notable as CHDs are the most prevalent of birth defects [54] occurring in 8–10 of every 1000 live births in the US [55,56,57,58]. This is one of the first studies to evaluate subsequent maternal health in this population, which provides new insights into disrupted metabolic pathways associated with CHD-affected pregnancies.

Specifically, among mothers with HLHS-affected pregnancies, we observed higher levels of PC (34:1) (also known as PC aa C34:1) and PC-O (40:4) (phosphatidylcholine with an alkyl ether substituent, also known as PC ae C40:4) when compared to mothers of controls. Glycerophospholipids are lipids composed of glycerol, two fatty acids, phosphate, and an amino alcohol. They are ubiquitous in cell membranes and are involved in the metabolism of cell signaling and permeability as well as maintaining the structural integrity of cell membranes.

Phosphatidylcholine (PC) is the most abundant phospholipid in mammalian cell membranes comprising ~50% of the total phospholipid mass of most cells and their organelles depending on the cell types [59]. Changes in the PC profiles were linked with metabolic disorders, such as atherosclerosis, insulin resistance, and obesity [60]. Additionally, PC (34:1) has been identified as part lipid profiles in mother-infant pairs, with levels being more abundant in mothers than their infants [61].

Levels of PC (35:0), a glycerophospholipid lecithin, is the only metabolite consistently lower among women who had HLHS- and ToF-affected pregnancies (Figure 4). Lecithin can be found in many foods, including soybeans and egg yolks, and has been shown to reduce hypercholesteremia and atherosclerosis [62]. These changes could point to novel interventions (e.g., dietary supplementation) among women who gave birth to children with CHDs.

Lysophosphatidylcholine (LPC), also known as lysolecithins, has received increased attention in relation to cardiovascular diseases. It is a class of lipid biomolecule derived by the cleaving of PC through the phospholipase A2 (PLA2) enzyme [63] or through the conversion of fatty acids to free cholesterol via lecithin-cholesterol acyltransferase (LCAT) [64]. Plasma LPCs have been shown to be inversely associated with cardiovascular disease [65,66,67] but have not been directly implicated in embryonic cardiac development.

In our study, we observed lower levels of LPC (18:2) (also known as lysoPC a C18:2) and LPC (20:3) (also known as lysoPC a C20:3) among women with ToF offspring compared with the controls (Table 3 and Figure 6). Bahado-Singh et al. also reported lower levels of maternal serum LPC (18:2) and LPC (20:3) among mothers of children with CHDs when compared to the mothers of controls [10]. In their study, 17 out of 27 total CHD cases were ToF defects, including double outlet right ventricle (DORV)/ToF (two cases), ToF alone (nine cases), ToF/mitral stenosis (one case), and ToF/pulmonary atresia (five cases) [10].

Complex changes occur in lipid profiles during pregnancy. The TG, total cholesterol (TC), low-density lipoprotein cholesterol (LDL-C), and high-density lipoprotein cholesterol (HDL-C) all increase significantly by the third month or at the end of the first trimester in response to elevated estrogen levels and insulin resistance [68]. The increase in the lipid metabolism during pregnancy aids the nutrient and energy sources for the fetus. In our study, lower levels of TG (44:1) and TG (46:2) were observed among women with ToF offspring than controls (Table 3 and Figure 6), reflecting the altered metabolic health of CHD-affected pregnancies.

Pathway analyses revealed changes in the D-glutamine and D-glutamate metabolism when comparing HLHS with controls and the glutamatergic synapse when comparing ToF with controls. Glutamine is the most abundant amino acid in humans. It plays a critical role in stimulating blood flow and increases blood fluidity by the synthesis of nitric oxide [69,70]. It can be transported into cells and further metabolized to glutamate by the mitochondrial enzyme glutaminase.

There are strong lines of evidence demonstrating that fundamental role of glutamine in maintaining cardiovascular health [71], and altered D-glutamine and D-glutamate metabolisms were identified as significantly impacted pathways among adults with congenital heart disease when compared with healthy controls [72]. Furthermore, glutamine supplements have also been shown to improve the cardiac function of patients with chronic heart failure [73]. We were unable to find any reports published in English in PubMed of glutamine among women with CHF-affected pregnancies. Further study is needed to confirm with the potential of using glutamine as dietary supplement to improve maternal health.

Our study must be considered in the light of certain limitations. The main limitation from our study was the small sample size for each CHD-subtype, which limits the statistical precision in our models. The time of blood collection was not available in this study, and might contribute to the variations in the metabolome.

Although we found no significant differences between the groups regarding pre-existing metabolic syndromes during pregnancy, such as high blood pressure and Type II Diabetes, pre-pregnancy plasma samples were not available, and thus the differences observed may be explained by other factors than CHD-affected pregnancy, such as environmental exposures [74,75], pre-existing metabolic conditions, and lifestyle factors before or during pregnancy [76,77,78].

In this pilot study, we demonstrated the feasibility of using targeted metabolomics to identify biomarkers for cardiometabolic health among mothers with a CHD-affected pregnancy. Since CHD is a rare condition with various subtypes, it is challenging to obtain a large patient base in a single biomarker study [10,79,80]. Nevertheless, our assessment adds to a growing body of literature indicating that mothers of children with birth defects may be at risk for adverse health conditions. Future studies collaborating with other birth defect centers are needed to confirm our results and further characterize these affected metabolic pathways.

## Figures and Tables

**Figure 1 metabolites-12-00100-f001:**
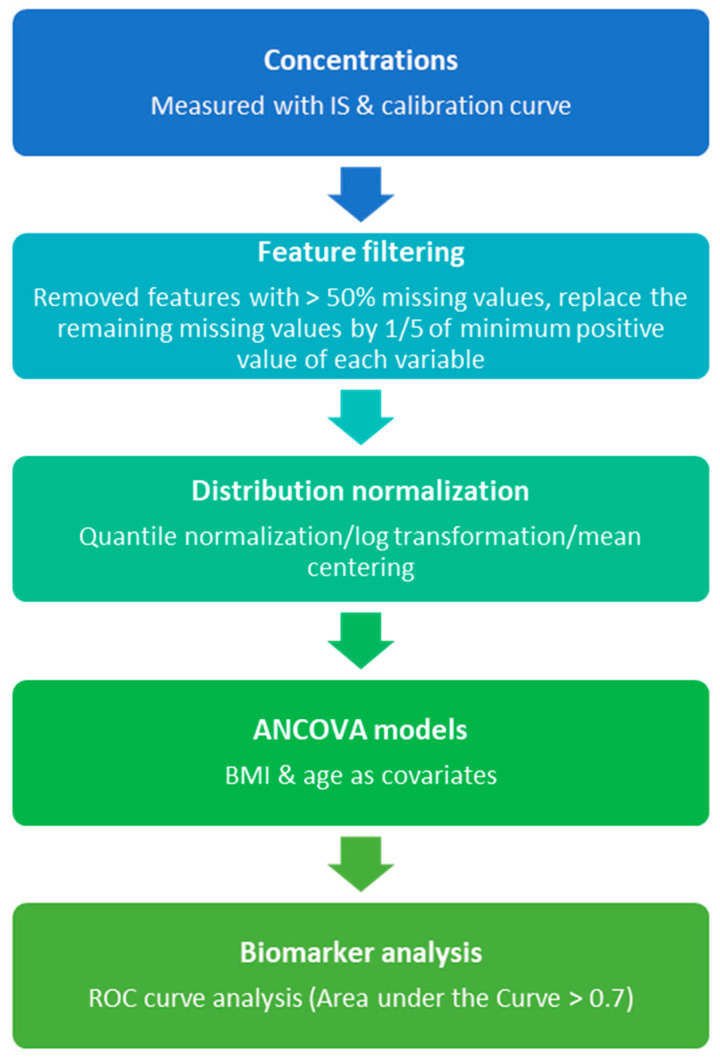
Metabolomics analysis workflow.

**Figure 2 metabolites-12-00100-f002:**
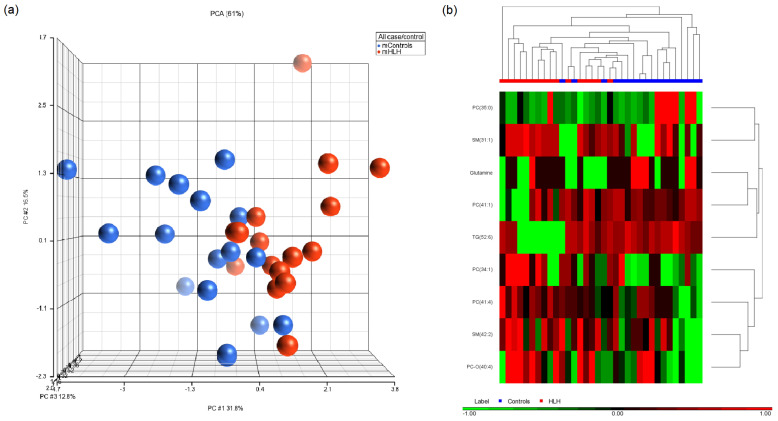
Visualization of the significant metabolites (*p* < 0.05) in (**a**) principle component analysis and (**b**) hierarchical clustering distinguishing between mControl vs. mHLHS.

**Figure 3 metabolites-12-00100-f003:**
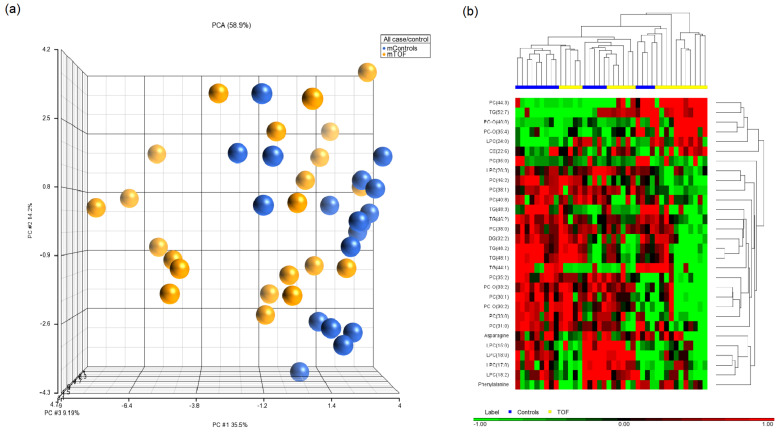
Visualization of the significant metabolites (*p* < 0.05) in (**a**) principle component analysis and (**b**) hierarchical clustering distinguishing between mControls vs. mToF.

**Figure 4 metabolites-12-00100-f004:**
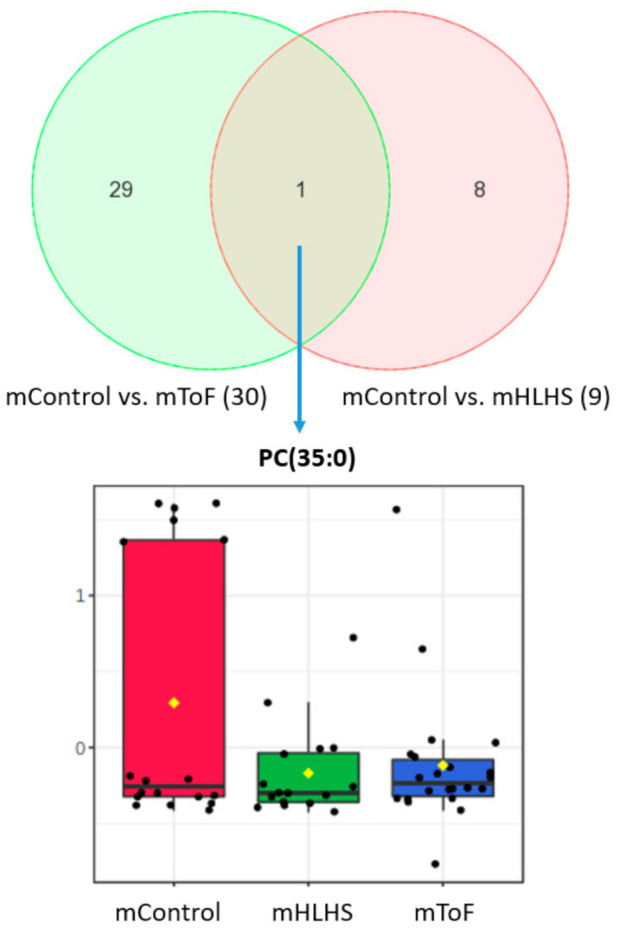
Box plot of significant metabolite overlapping between two comparisons (mControls vs. mTOF and mControl vs. mHLHS). Levels of PC (35:0), a glycerophospholipid lecithin, were significantly higher in the mControl compared to mHLHS and mToF.

**Figure 5 metabolites-12-00100-f005:**
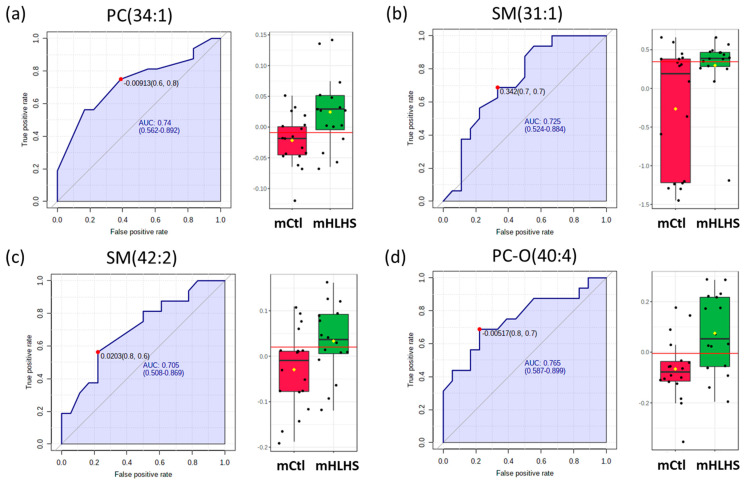
Biomarker analysis for the predictive values metabolites for mHLHS. The sensitivity (true positive rate) is on the *y*-axis, and the specificity (one minus the false positive rate) is on the *x*-axis, with the area under the curve (AUC) > 0.70 on four metabolites, including (**a**) PC (34:1), (**b**) SM (31:1), (**c**) SM (42:2) and (**d**) PC-O (40:4). A horizontal line in red in the box plots indicates the optimal cutoff of between two groups.

**Figure 6 metabolites-12-00100-f006:**
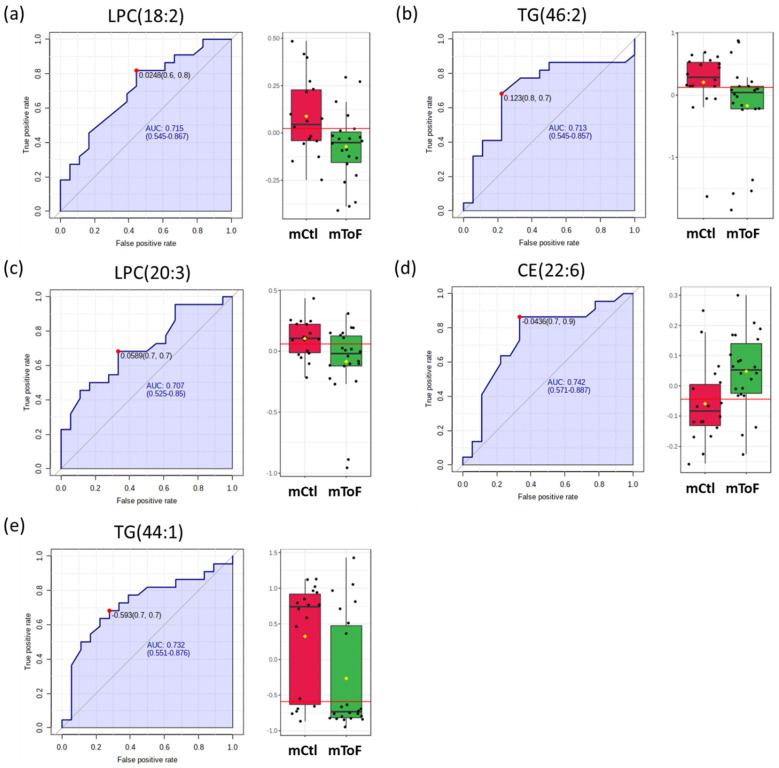
Biomarker analysis for the predictive values metabolites for mToF. The sensitivity (true positive rate) is on the *y*-axis, and the specificity (one minus the false positive rate) is on the *x*-axis, with the area under the curve (AUC) > 0.70 on six metabolites, including (**a**) LPC (18:2), (**b**) TG (46:2), (**c**) LPC (20:3), (**d**) CE (22:6), and (**e**) TG (44:1). A horizontal line in red in the box plots indicates the optimal cutoff of between two groups.

**Figure 7 metabolites-12-00100-f007:**
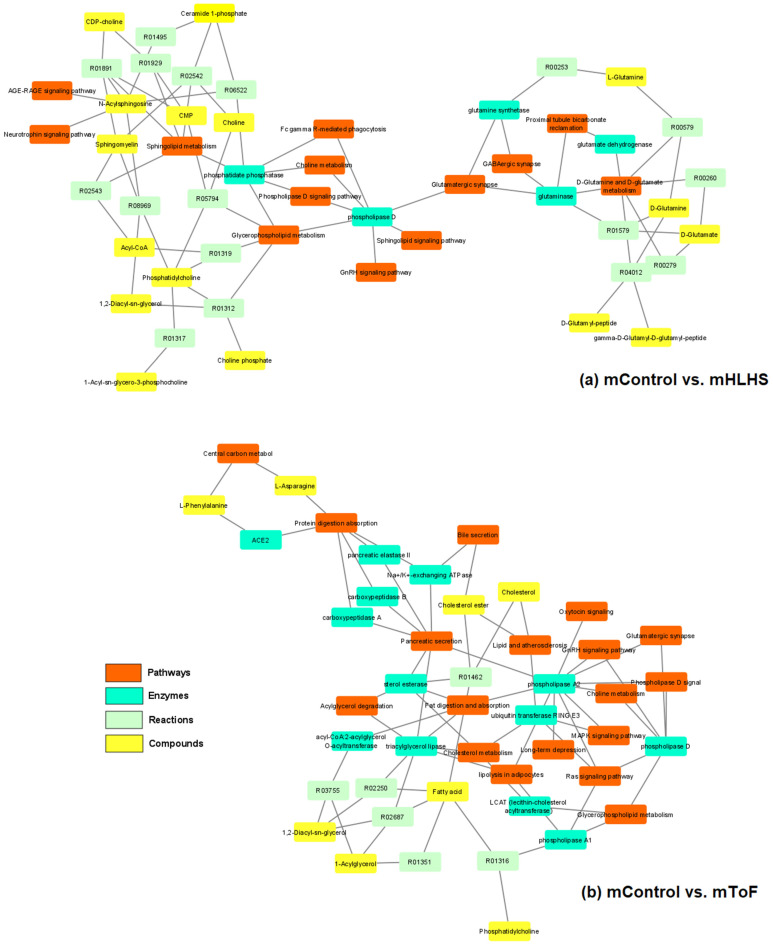
Metabolic pathways associated with (**a**) mHLHS and (**b**) mToF pregnancies when comparing with normal controls. The depicted networks revealed pathways (orange nodes), enzymes (light blue nodes), reactions (light green nodes), and metabolites (yellow nodes) that were involved in the pathways affected (*p* < 0.01).

**Table 1 metabolites-12-00100-t001:** Maternal demographic factors among the CHD control and cases from the Arkansas site of NBDPS (1997–2011) included in the study.

	mControl ^1^ (18)	mHLHS ^2^ (16)	mToF ^3^ (22)	*p* *
	Mean ± Std/Counts			
Pre-pregnancy BMI ^4^	25.9 ± 6.6	24.5 ± 4.6	26.6 ± 7.6	0.6
Post-partum BMI	31.6 ± 6.0	28.6 ± 3.7	31.0 ± 6.8	0.3
BMI change	5.8 ± 2.7	4.2 ± 2.4	4.4 ± 2.2	0.1
Weight change (kg)	15.4 ± 8.0	11.3 ± 6.8	11.6 ± 5.9	0.1
Maternal age at delivery	25.8 ± 5.6	27.3 ± 5.5	26.6 ± 5.6	0.7
Maternal age at EDD ^5^	25.9 ± 5.7	27.3 ± 5.5	26.6 ± 5.6	0.8
Maternal age at conception	25.2 ± 5.7	26.8 ± 5.5	26.0 ± 5.7	0.7
Maternal age at delivery	25.8 ± 5.6	27.3 ± 5.5	26.6 ± 5.6	0.7
Race				0.6
Non-Hispanic white	17	14	20	
Non-Hispanic black	1	1	2	
Asian/Pacific Islander	0	1	0	
Alcohol consumption				0.2
Yes (B1-M3) **	10	6	6	
No	8	10	16	
Maternal smoking				0.9
Heavy smokers or 15+ cigs per day	2	1	2	
Medium smokers or 5–14 cigs per day	3	3	2	
Light smokers or ≤1–4 cigs per day	1	2	3	
No smoking	12	10	15	
High blood pressure during pregnancy				0.7
Yes	1	3	5	
No	1	1	1	
No answer	16	12	16	
High blood pressure medicine use				0.3
Yes	0	1	0	
No	1	3	6	
No answer	17	12	16	
Type II Diabetes				0.9
Yes	2	2	2	
No	16	14	20	

* *p*-values represent differences between groups. Continuous variables were evaluated by ANOVA, and chi square (*X*^2^) tests were used to investigate the differences in distributions of categorical variables. ^1^ mControl: mothers of unaffected offspring; ^2^ mHLHS: mothers of offspring with hypoplastic left heart syndrome; ^3^ mToF: mothers of offspring with tetralogy of Fallot; ^4^ BMI: body mass index; ^5^ EDD: estimated due date. ** One month before conception to end of first trimester.

**Table 2 metabolites-12-00100-t002:** Significant metabolites between the mothers of control infants (mControl) compared to the mothers of infants with hypoplastic left heart syndrome (mHLHS), Arkansas site of NBDPS, 1997–2011.

Metabolites	mControl	mHLHS	mControl vs. mHLHS (mControl/mHLHS)		
	Mean ± Std (μM)		*p* *	FC *	Trend
PC (34:1)	149.56 ± 29.13	157 ± 36.54	0.014	−1.03	Controls down
TG (52:6)	2.98 ± 1.44	2.81 ± 2.03	0.016	2.02	Controls up
PC (41:4)	1.65 ± 0.67	1.90 ± 0.62	0.020	−1.27	Controls down
SM (31:1)	0.49 ± 0.26	0.48 ± 0.25	0.023	−1.43	Controls down
SM (42:2)	39.98 ± 7.62	41.38 ± 6.63	0.027	−1.07	Controls down
PC (41:1)	0.66 ± 0.36	0.40 ± 0.36	0.029	1.38	Controls up
Glutamine	548 ± 73.95	484.10 ± 63.81	0.046	1.04	Controls up
PC (35:0)	0.12 ± 0.18	0.008 ± 0.009	0.046	1.39	Controls up
PC-O (40:4)	0.95 ± 0.28	1.20 ± 0.49	0.049	−1.10	Controls down

* FC = fold change. Raw *p*-values and FC represent differences between groups.

**Table 3 metabolites-12-00100-t003:** Significant metabolites between the mothers of control infants (mControl) compared to the mothers of infants with Tetralogy of Fallot (mToF), Arkansas site of NBDPS, 1997–2011.

Metabolites	mControl	mToF	mControl vs. mToF (mControl/mToF)		
	Mean ± Std (μM)		*p* *	FC *	Trend
LPC (18:2)	14.62 ± 7.17	9.87 ± 3.76	0.004	1.15	Controls up
Asparagine	45.87 ± 7.47	39.85 ± 10.07	0.006	1.08	Controls up
PC (31:0)	0.54 ± 0.22	0.37 ± 0.18	0.010	1.16	Controls up
LPC (24:0)	0.105 ± 0.11	0.111 ± 0.10	0.014	−2.11	Controls down
PC (40:8)	6.96 ± 5.94	3.71 ± 4.23	0.014	1.36	Controls up
PC-O (40:0)	0.006 ± 0.003	0.004 ± 0.003	0.017	−2.54	Controls down
TG (52:7)	1.20 ± 0.82	1.86 ± 0.95	0.017	−2.33	Controls down
TG (44:1)	1.86 ± 1.59	0.78 ± 1.39	0.018	1.55	Controls up
LPC (20:3)	2.42 ± 1.11	1.54 ± 0.73	0.020	1.20	Controls up
PC (38:1)	0.59 ± 0.22	0.42 ± 0.23	0.021	1.17	Controls up
LPC (15:0)	0.59 ± 0.19	0.44 ± 0.13	0.025	1.08	Controls up
PC-O (30:2)	0.008 ± 0.001	0.007 ± 0.001	0.025	1.07	Controls up
PC (38:0)	0.66 ± 0.31	0.46 ± 0.31	0.026	1.20	Controls up
PC-O (33:2)	0.005 ± 0.001	0.004 ± 0.001	0.028	1.08	Controls up
PC-O (35:4)	0.29 ± 0.35	0.15 ± 0.05	0.032	−1.82	Controls down
Phenylalanine	65.61 ± 13.5	52.38 ± 8.63	0.032	1.04	Controls up
PC (33:0)	0.85 ± 0.25	0.69 ± 0.17	0.034	1.05	Controls up
TG (48:3)	5.78 ± 6.72	2.60 ± 5.63	0.037	1.59	Controls up
PC (44:3)	1.65 ± 2.7	4.89 ± 5.42	0.038	−1.82	Controls down
LPC (18:0)	22.04 ± 8.58	17.62 ± 6.67	0.040	1.09	Controls up
LPC (17:0)	1.43 ± 0.45	1.144 ± 0.44	0.041	1.08	Controls up
PC (35:2)	8.02 ± 2.15	6.43 ± 1.57	0.041	1.06	Controls up
CE (22:6)	65.41 ± 23.51	75.38 ± 21.32	0.041	−1.08	Controls down
PC (35:0)	0.12 ± 0.18	0.03 ± 0.09	0.042	1.36	Controls up
PC (30:1)	0.009 ± 0.002	0.008 ± 0.001	0.045	1.07	Controls up
TG (48:2)	13.74 ± 8.14	9.71 ± 8.08	0.047	1.13	Controls up
TG (46:2)	3.72 ± 2.61	2.30 ± 3.28	0.047	1.36	Controls up
TG (48:1)	15.35 ± 9.04	10.49 ± 8.54	0.048	1.14	Controls up
DG (32:2)	0.77 ± 0.40	0.55 ± 0.37	0.048	1.12	Controls up
PC (46:2)	8.91 ± 4.99	6.26 ± 4.14	0.049	1.12	Controls up

***** FC = fold change. Raw *p*-values and FC represent differences between groups.

## Data Availability

The data presented in this study are available on request from the corresponding author. The data are not publicly available due to privacy reasons.

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
