# Peer review of "Metabolomics Signatures and Subsequent Maternal Health among Mothers with a Congenital Heart Defect-Affected Pregnancy"

_metabolites, 2022, doi:10.3390/metabo12020100_

Round 1
Reviewer 1 Report
The work described in this manuscript consists of a postpartum metabolomic analysis in plasma of mothers of children with a CHD (TOF and HLHS) . The authors suggest that such analysis can give clues to the cause of CHDs. The results show small but significant differences between CHD mothers and controls in 9 metabolites of HLHS mothers and 30 metabolites of TOF mothers. This is an interesting and novel approach to an important question, that is, what causes CHDs?
In my opinion, there are several important points that should be addressed to make the obtained results significant and relevant.
1) It has to be explained or a model should be presented on how a postpartum metabolic profile could explain CHDs of the children. A metabolic profile during pregnancy would be more adequate in finding potential causes of congenital defects of development if it has any role in its etiology. CHDs these days are readily detected during pregnancy by ultrasonography in routine practice and thus metabolic analysis of pregnant mothers with a fetus suffering of CHD is possible.
2) I am not sure that they rule out that the significant differences in metabolite levels could not be due to any of the sources of variation (Sup Fig 2). I think that this should be ruled out. As many of the significant metabolites are lipids, BMI, in particular, could be an important confounding factor ( show a PCA analysis with the BMIs?).
3) The quality of Figure 7 is insufficient and blurry. I have not been able to read any of the labels within the figure.
Author Response
Point 1: It has to be explained or a model should be presented on how a postpartum metabolic profile could explain CHDs of the children. A metabolic profile during pregnancy would be more adequate in finding potential causes of congenital defects of development if it has any role in its etiology. CHDs these days are readily detected during pregnancy by ultrasonography in routine practice and thus metabolic analysis of pregnant mothers with a fetus suffering of CHD is possible.
Response 1:
We apologize for not being clear. The study was to determine the metabolomics signatures on subsequent maternal health among mothers with a congenital heart defect-affected pregnancy. We have revise the title so that it is clearer and not misleading.
Point 2: I am not sure that they rule out that the significant differences in metabolite levels could not be due to any of the sources of variation (Sup Fig 2). I think that this should be ruled out. As many of the significant metabolites are lipids, BMI, in particular, could be an important confounding factor (show a PCA analysis with the BMIs?).
Response 2:
We appreciate the concern, and have addressed age and BMI as co-variates in the Analysis of covariance (ANCOVA) model adjusting for maternal age at delivery and BMI in “Metabolomics Data Analysis” under “Materials and Methods”. The F ratios for BMI were 1.35 in mHLHS and 1.56 in mToF comparing to 1 in random errors (see supplemental Figure 3a-b), and thus was not considered as an important confounding factor in the model.
Point 3: The quality of Figure 7 is insufficient and blurry. I have not been able to read any of the labels within the figure.
Response 3:
We apologize for this error, and have replaced figure 7 with a higher resolution file.

Reviewer 2 Report
In this study the authors explore the postpartum metabolomic profile of mothers with CHD-affected pregnancies, specifically HLHS and TOF, in comparison to mothers without CHD offspring. Using postpartum plasma samples collected as part of the National Birth Defects Prevention study, targeted metabolic analyses of 408 metabolites was conducted, controlling for maternal age and BMI. This identified 9 (or 24? – see comment 2 below) metabolites that were significantly different between HLHS mothers and controls and 30 (or 63?) between TOF mothers and controls, prior to correction for multiple testing. Following correction for multiple testing, no significant differences were observed.
The authors should be congratulated on an interesting and timely study that is well written and clearly presented. A few comments:
Major comment:
- This study is based on a single postpartum serum analysis. Without a pre-pregnancy serum sample to provide baseline metabolic measures between the mothers in the 3 groups, it is plausible that the differences observed in maternal metabolomic profiles between mothers of controls and CHD-affected pregnancies, may not be a result of the CHD-affected pregnancy but rather a contributor to having a CHD-affected pregnancy i.e. one could argue that these differences may have existed pre-pregnancy and could be a risk factor for having a baby with CHD. Are pre-pregnancy/early pregnancy serum samples available to conduct a baseline analysis in the three groups for comparison? If not, the authors should comment on this as a limitation to the study and caution should be exercised in the interpretation of the findings.
- In the abstract the authors report that there were 24 significant metabolites in HLHS cases and 63 in TOF cases but in the text in the Results section (line 154-167) and Tables 2 and 3, the authors report 9 significant metabolites in HLHS cases and 30 in TOF? Please clarify.
- The authors briefly note that none of the metabolites were significant following adjusting for multiple comparisons in the results, but this should also be discussed in relation to the interpretation of findings as part of the discussion, as theoretically (and from a statistical perspective), the analysis yielded no significant results
- Mention of non-significance following correction for multiple testing is provided in the text but this should also be highlighted in Table 2 and Table 3 to make this clear to the reader (could also be as a footnote to the table)
- Line 167: Incorrect table number provided when discussing mTOF results, should refer to Table 3 (not Table 2)
Author Response
Point 1: This study is based on a single postpartum serum analysis. Without a pre-pregnancy serum sample to provide baseline metabolic measures between the mothers in the 3 groups, it is plausible that the differences observed in maternal metabolomic profiles between mothers of controls and CHD-affected pregnancies, may not be a result of the CHD-affected pregnancy but rather a contributor to having a CHD-affected pregnancy i.e. one could argue that these differences may have existed pre-pregnancy and could be a risk factor for having a baby with CHD. Are pre-pregnancy/early pregnancy serum samples available to conduct a baseline analysis in the three groups for comparison? If not, the authors should comment on this as a limitation to the study and caution should be exercised in the interpretation of the findings.
Response 1:
We apologize for this error, and have replaced figure 7 with a higher resolution file.
Point 2: In the abstract the authors report that there were 24 significant metabolites in HLHS cases and 63 in TOF cases but in the text in the Results section (line 154-167) and Tables 2 and 3, the authors report 9 significant metabolites in HLHS cases and 30 in TOF? Please clarify.
Response 2:
We sincerely apologize for this error, and have revised the abstract for the correct numbers (9 in mHLHS and 30 in mTOF).
Point 3: The authors briefly note that none of the metabolites were significant following adjusting for multiple comparisons in the results, but this should also be discussed in relation to the interpretation of findings as part of the discussion, as theoretically (and from a statistical perspective), the analysis yielded no significant results.
Response 3:
We appreciate this comment, and did mention in the abstract that this is a pilot study, and in the last paragraph in the discussion that “In this pilot study we demonstrated the feasibility of using targeted metabolomics to identify biomarkers for cardiometabolic health among mothers with a CHD-affected pregnancy”. We have added “A significance level of P < .05 was used to define statistical significance” under the Metabolomics Data Analysis, removed “After adjusting for multiple comparisons, there were no significant differences across groups” in the results to avoid confusion, and also addressed in the discussion that “Since CHD is a rare condition with varies subtypes, it is challenging to obtain large patient base in a single biomarker study as most studies were (Bahado-Singh et al., 2014; Clausen et al., 2020; Neves et al., 2016)”; and “Future studies collaborating with other birth defect centers are needed”.
Point 4: Mention of non-significance following correction for multiple testing is provided in the text but this should also be highlighted in Table 2 and Table 3 to make this clear to the reader (could also be as a footnote to the table).
Response 4:
We appreciate this comment, and have revised the footnote of Table 2 & 3 as “raw p-values”.
Point 5: Line 167: Incorrect table number provided when discussing mTOF results, should refer to Table 3 (not Table 2).
Response 5:
We appreciate the reminder, and have revised the error.

Reviewer 3 Report
In this manuscript the authors report on a pilot study in which they have analyzed the post-partum metabolic profiles of women who gave birth to children with congenital heart defects (case mothers) and compared these profiles with the profiles obtained from women who gave birth to morphologically normal children (control mothers, n=18). Two groups of case mothers were studied, (1) those who had offsprings with Tetralogy of Fallot (ToF, n=22); and (2) those who had offsprings with hypoplastic left heart syndrome (HLHS, n=16). It is said that the case groups were selected based on "phenotypic severity"?? (whatever this means). The study identified several metabolites whose concentrations showed statistically significant differences to the control group. The authors conclude that pregnancies with fetuses with congenital heart defects (CHD) may have adversely changed the post-partum metabolic profile of the affected mothers and, thereby, may bring the mothers at risk for adverse health conditions.
I have several problems with the manuscript:
(1) In the "Introduction", I miss references to previous studies, which have analyzed the metabolic profiles (serum, urine) of pregnant women carrying fetuses with CHD. These references are important since they demonstrate that pregnancies affected by CHD are associated with altered metabolic profiles of the mothers during pregnancy (pre-partum). These studies, therefore, provide an important rational for conducting the present post-partum study (e.g. Friedman et al. 2021, J Matern Fetal Neonatal Med:1-8; Troisi et al. 2021, Prenat Diagn 41:743-753; Xie et al. 2019, Biomed Res Int:1905416; Bahada-Singh et al. 2014, Am J Obst Gynecol 211:240.e1-240.e14 (the latter study is mentioned in the discussion of this MS but is missed in the list of references)).
(2) In "Materials and Methods" I miss some information on the study design: (i) What is the time-relation between the date of birth and the date of post-partum venipuncture of the mother? (ii) Where the blood samples collected at a specific day time? (iii) Why did you collect only a single blood sample from each mother? (iv) What are the reasons for choosing the groups of ToF and HLHS? (v) Is it correct that you selected the participants for each group on the basis of phenotypic severity of the cases, e.g. inclusion of HLHS with severe clinical presentation vs exclusion of HLHS with mild clinical presentation?
(3) The authors speculate that the abnormal post-partum metabolic profiles of the mothers, who gave birth to children with ToF or HLHS, have resulted from the pregnancies affected by CHD. In view of the fact that many cases of CHD have an inherited genetic base, I miss discussion of the possibility that the abnormal metabolic profiles of these mothers may reflect their genetic conditions.
(4, minor comment) In the discussion (page 11, lines 264-266), reference is given to a study conducted by Bahada-Singh et al.. This study is missed in the list of references. Did you mean the following study: Bahada-Singh et al. 2014, Am J Obst Gynecol 211:240.e1-240.e14 ?
Author Response
Point 1: In the "Introduction", I miss references to previous studies, which have analyzed the metabolic profiles (serum, urine) of pregnant women carrying fetuses with CHD. These references are important since they demonstrate that pregnancies affected by CHD are associated with altered metabolic profiles of the mothers during pregnancy (pre-partum). These studies, therefore, provide an important rational for conducting the present post-partum study (e.g. Friedman et al. 2021, J Matern Fetal Neonatal Med:1-8; Troisi et al. 2021, Prenat Diagn 41:743-753; Xie et al. 2019, Biomed Res Int:1905416; Bahada-Singh et al. 2014, Am J Obst Gynecol 211:240.e1-240.e14 (the latter study is mentioned in the discussion of this MS but is missed in the list of references)).
Response 1:
We appreciate this useful comment, and have added the following to introduction: “Women with pregnancies affected by CHD have been reported with altered metabolic profiles and were associated with dysregulated metabolic pathways.1-4 These markers were often utilized as predictive markers for fetuses with congenital heart defects”. We have also added the citation for Bahada-Singh et al. 2014 in the discussion.
Point 2: In "Materials and Methods" I miss some information on the study design: (i) What is the time-relation between the date of birth and the date of post-partum venipuncture of the mother? (ii) Where the blood samples collected at a specific day time? (iii) Why did you collect only a single blood sample from each mother? (iv) What are the reasons for choosing the groups of ToF and HLHS? (v) Is it correct that you selected the participants for each group on the basis of phenotypic severity of the cases, e.g. inclusion of HLHS with severe clinical presentation vs exclusion of HLHS with mild clinical presentation?
Response 2:
Please see our response below:
(i) What is the time-relation between the date of birth and the date of post-partum venipuncture of the mother?
We apologize for missing that information. After receiving written informed consent, blood samples were obtained at least 6 weeks postpartum by the study nurse. It has been added to the Study Design & Study Participants in the Materials and Methods.
(ii) Where the blood samples collected at a specific day time?
During home visits, the nurse obtained written consent, performed venipuncture to obtain blood. Unfortunately we don’t have the info on time of collection, and has been addressed in the weakness in the discussion: “Time of blood collection was not available, and might contribute to the variations in the metabolome”.
(iii) Why did you collect only a single blood sample from each mother?
The study was originally proposed to identify genetic variants, such as single nucleotide polymorphism (SNPs) and copy number variants (CNVs) that are associated with CHDs. Therefore, only one blood sample up to 30mL was collected for each women.
(iv) What are the reasons for choosing the groups of ToF and HLHS?
mHLHS and mTOF were chosen because TOF and HLHS cases are presented with more severe phenotypes, which might be indicative of more severe maternal phenotypes. Also, they might be a bit more homogeneous in their presentation than septal defects, which are much more common and variable in presentation.
(v) Is it correct that you selected the participants for each group on the basis of phenotypic severity of the cases, e.g. inclusion of HLHS with severe clinical presentation vs exclusion of HLHS with mild clinical presentation
No. Hypoplastic left heart syndrome (HLHS) itself is a severe congenital heart defect in which the left side of the heart is underdeveloped. There is no “mild” clinical presentation of HLHS.
Point 3: The authors speculate that the abnormal post-partum metabolic profiles of the mothers, who gave birth to children with ToF or HLHS, have resulted from the pregnancies affected by CHD. In view of the fact that many cases of CHD have an inherited genetic base, I miss discussion of the possibility that the abnormal metabolic profiles of these mothers may reflect their genetic conditions.
Response 3:
We appreciate this comment, however, according to the Scientific Statement from the American Heart Association5: “the largest genetic study of congenital heart defect suggested that 8% and 2% of cases are attributable to de novo autosomal dominant and inherited autosomal recessive variation, respectively.6 Environmental causes are identifiable in 2% of congenital HD cases. The unexplained remainder of congenital HD is presumed to be multifactorial (oligogenetic or some combination of genetic and environmental factors)7”. Considering the small percentage of the cases from genetic inheritance, and also the small sample size in this pilot study, we did not include that discussion.
Point 4: (minor comment) In the discussion (page 11, lines 264-266), reference is given to a study conducted by Bahada-Singh et al.. This study is missed in the list of references. Did you mean the following study: Bahada-Singh et al. 2014, Am J Obst Gynecol 211:240.e1-240.e14 ?
Response 4:
Yes, it was the source of the message, and we apologize for missing this citation. We have included it in the reference.

Reviewer 4 Report
The study by Hsu et al. used a Metabolomics approach to study the impact CHD-affected pregnancies on subsequent maternal health. The authors report that 24 metabolites were significant (p<0.05) in hypoplastic left heart syndrome (HLHS) compared to control, where as 63 metabolites were significant in tetralogy of Fallot (ToF) mothers compared to control. Overall the study is fairly comprehensive and detailed, i have some comments and questions about the data.
1) Authors have indicated that the samples were collected over a period of many years, it would be more helpful to indicate how quality of samples was maintained over the years without any possible degradation until the time of mass spectrometer analysis, given metabolites mentioned in the study can have varied stability.
Also can authors describe if there is any mass spectrometer variation or quantitative difference specifically in control samples based on year of sample collection.
2) Authors have indicated that metabolites present below limit of detection (LOD) were imputed by half of the minimum peak intensity of the entire dataset. Please indicate what percentage of the entire dataset was below LOD and imputed.
3) Authors have not described detailed LC-MS/MS methods, i would recommend adding details about LC conditions, MS/MS acquisition parameters and software used for raw mass spectrometer data analysis.
Author Response
Point 1: Authors have indicated that the samples were collected over a period of many years, it would be more helpful to indicate how quality of samples was maintained over the years without any possible degradation until the time of mass spectrometer analysis, given metabolites mentioned in the study can have varied stability.
Response 1:
We appreciate this comment, and have added the description “Samples were stored in locked freezers, which are only accessible to authorized study personnel and facility management. The temperature of the freezer were continuously monitored by an electronic monitoring system and routinely monitored by the biorepository personnel to ensure the integrity of the samples” under Study Design & Study Participants in the Materials and Methods.
Point 2: Also can authors describe if there is any mass spectrometer variation or quantitative difference specifically in control samples based on year of sample collection.
Response 2:
We appreciate this comment. Controls were samples from mother of infants without birth defects who were randomly selected from all birth certificates given to the Vital Statistics Division of the Arkansas Department of Health. The year of sample collection from 18 control samples were ranging from year 1999 to 2008, and did not show any mass spectrometer variations base on the PCA plot shown below. This info as well as the PCA from all samples plotted with the infant year of birth is now added to supplemental figure 2.
Point 3: Authors have indicated that metabolites present below limit of detection (LOD) were imputed by half of the minimum peak intensity of the entire dataset. Please indicate what percentage of the entire dataset was below LOD and imputed.
Response 3:
We thank the reviewer for the suggestion. We have added Supplemental Table 2 presenting a summary of data processing results including the number of metabolites imputed due to below LOD value, and have added “Summary of data processing results including the number of metabolites imputed due to missing or below LOD value can be seen in Table S2” under the “Targeted metabolomics profiling”.
Point 4: Authors have not described detailed LC-MS/MS methods, i would recommend adding details about LC conditions, MS/MS acquisition parameters and software used for raw mass spectrometer data analysis.
Response 4:
We apologize for missing this information, and have added detailed LC-MS/MS methods to the manuscript under “Targeted Metabolite Profiling” in the Materials and Methods.

Round 2
Reviewer 1 Report
Changes introduced have improved the manuscript.
Author Response
Thank you for the comment from the reviewer!
Reviewer 3 Report
Authors have successfully addressed all of my previous comments except one; this is point 3: "The authors speculate that the abnormal post-partum metabolic profiles of the mothers, who gave birth to children with ToF or HLHS, have resulted from the pregnancies affected by CHD. In view of the fact that many cases of CHD have an inherited genetic base, I miss discussion of the possibility that the abnormal metabolic profiles of these mothers may reflect their genetic conditions." In your response you refer to a Scientific Statement from the American Heart Association and conclude that "Considering the small percentage of the cases from genetic inheritance, and also the small sample size in this pilot study, we did not include that discussion."
Comment: I wonder about this reply. Current review articles on the genetic base of CHDs clearly show that genetic etiologies contribute to an estimated 90% of CHD cases (see for example Diab et al. 2021, Genes 12(7) 1020). It is furthermore known that environmental factors, such as metabolic illnesses of the mother (diabetes mellitus, hypercholesterolemia), play a fundamental role in some types of CHDs. I, therefore, strongly suggest discussing the possibility that the post-partum metabolic profiles of the mothers of children with ToF and HLHS may be explained by other factors than CHD-affected pregnancy (e.g. genetic condition of the mother, preconceptional metabolic illness of the mother).
Author Response
We thank the reviewer for their comment; however, we respectfully disagree with this point. As stated in the review article by Pierpont et al, “Genetic Basis for Congenital Heart Disease: Revisited: A Scientific Statement From the American Heart Association”,1 and further endorsed by the American Academy of Pediatrics,2 “Single-gene disorders are found in 3% to 5%, gross chromosomal anomalies/aneuploidy in 8% to 10%, and pathogenic CNVs in 3% to 25% of those with congenital HD as part of a syndrome, and in 3% to 10% among those with isolated congenital HD. The largest genetic study of congenital HD with NGS suggested that 8% and 2% of cases are attributable to de novo autosomal dominant and inherited autosomal recessive variation, respectively.3 Environmental causes are identifiable in 2% of congenital HD cases. The unexplained remainder of congenital HD is presumed to be multifactorial (oligogenetic or some combination of genetic and environmental factors).”
The paper published by Diab et al.,4 stated that: “Despite exhaustive efforts by many groups, the causative genetic mechanisms behind CHD remain poorly understood and ~55% of CHD patients lack a genetic diagnosis”. Therefore, we want to be cautious about their conclusion in the abstract on “Genetic etiologies contribute to an estimated 90% of CHD cases”. Furthermore, the genetics studies reported were conducted for the infants and not the mothers, who were the focus of our study.
Therefore, we feel that the original statement in the manuscript is well supported by the published literature. We did, however, revise the discussion on the limitation of the study: “Although no significant difference between the groups on pre-existing metabolic syndromes during pregnancy such as high blood pressure and Type II Diabetes, pre-pregnancy plasma samples were not available and thus the differences observed may be explained by other factors than CHD-affected pregnancy such as environmental exposures,5, 6 pre-existing metabolic conditions and lifestyle factors before or during pregnancy7-9”.
References cited:
- Pierpont ME, Brueckner M, Chung WK, et al. Genetic Basis for Congenital Heart Disease: Revisited: A Scientific Statement From the American Heart Association. Circulation. 2018;138: e653-e711.
- Correction to: Genetic Basis for Congenital Heart Disease: Revisited: A Scientific Statement From the American Heart Association. Circulation. 2018;138: e713.
- Jin SC, Homsy J, Zaidi S, et al. Contribution of rare inherited and de novo variants in 2,871 congenital heart disease probands. Nat Genet. 2017;49: 1593-1601.
- Diab NS, Barish S, Dong W, et al. Molecular Genetics and Complex Inheritance of Congenital Heart Disease. Genes (Basel). 2021;12.
- Gong W, Liang Q, Zheng D, Zhong R, Wen Y, Wang X. Congenital heart defects of fetus after maternal exposure to organic and inorganic environmental factors: a cohort study. Oncotarget. 2017;8: 100717-100723.
- Peyvandi S, Baer RJ, Chambers CD, et al. Environmental and Socioeconomic Factors Influence the Live-Born Incidence of Congenital Heart Disease: A Population-Based Study in California. J Am Heart Assoc. 2020;9: e015255.
- Helle E, Priest JR. Maternal Obesity and Diabetes Mellitus as Risk Factors for Congenital Heart Disease in the Offspring. J Am Heart Assoc. 2020;9: e011541.
- Oyen N, Diaz LJ, Leirgul E, et al. Prepregnancy Diabetes and Offspring Risk of Congenital Heart Disease: A Nationwide Cohort Study. Circulation. 2016;133: 2243-2253.
- Feng Y, Yu D, Yang L, et al. Maternal lifestyle factors in pregnancy and congenital heart defects in offspring: review of the current evidence. Ital J Pediatr. 2014;40: 85.
